

# Impact of spatial resolution on CMIP6-driven Mediterranean climate simulations: a focus on precipitation distribution over Italy

Maria Vittoria Struglia[1,2], Alessandro Anav[1,2], Marta Antonelli[1], Sandro Calmanti[1,2], Franco Catalano[1,2], Alessandro Dell'Aquila[1], Emanuela Pichelli[1,2], Giovanna Pisacane[1]

[1]ENEA  Italian National Agency for New Technologies, Energy and Sustainable Economic Development, Rome, 00123, Italy
[2]ICSC Italian Research Center on High-Performance Computing, Big Data and Quantum Computing, Casalecchio di Reno (BO), 40033, Italy

*Correspondence to*: Maria Vittoria Struglia (mariavittoria.struglia@enea.it)

**Abstract.** We present the results of downscaling CMIP6 global climate projections to local scales for the Mediterranean and Italian regions, aiming to produce high-resolution climate information for the assessment of climate change signals, with a focus on precipitation extreme events. We performed hindcast (i.e. ERA5-driven) and historical simulations (driven by the MPI-ESM1-2-HR model) to simulate the present (1980-2014) and future (2015-2100) climate under three different emission scenarios (SSP1-2.6, SSP2-4.5, SSP5-8.5).

For each experiment, a double nesting approach is adopted to dynamically downscale global data to the regional domain of interest, firstly over the Europe (EURO) CORDEX domain, at a spatial resolution of 15 km, and then further refined (second nesting) over Italy and north-western Mediterranean, at a resolution of 5 km, i.e. in the so-called gray-zone (5-10 km), close to the convection permitting (CP) limit. Besides validating the experimental protocol, this work potentially questions the need for climate simulations to always resort to deep-convection parameterizations when spatial refinement is increased up to the

limit of the CP scale, yet convective processes are still not explicitly resolved. Analyses of the most relevant Essential Climate Variables (ECVs) are presented, with a focus on the spatial distribution of precipitation, its probability density function, and the statistics of extreme events, for both current climate and far-end scenarios. By the end of the century for all the scenarios and seasons there is a projected general warming along with an intensification of the hydrological cycle over most of the continental EU and mean precipitation reduction over the Mediterranean region accompanied, over Italian Peninsula, by a

strong increase in the intensity of extreme precipitation events, particularly relevant for the SSP5-8.5 scenario during autumn.

## 1 Introduction

In recent years, the availability of increasingly powerful computational resources has pushed regional modeling techniques to finer and finer scales, with demonstrated added value in comparison to coarser resolution of global models, especially in complex morphology regions (e.g., Torma et al. 2015).



Climate studies have benefited from such technological advances and regional climate projections have achieved the spatial and temporal resolution needed to assess the local impacts of climate change and climate-related risks and to support adaptation and mitigations policies (Giorgi et al., 2009, Torma et al., 2015, Giorgi et al., 2022). This represents a substantial breakthrough for the Mediterranean region, a climate hotspot characterized by a strongly heterogeneous morphology (a semi-closed basin with high and complex mountainous surroundings), which inherently demands for high-resolution analyses. The region is, in

fact, critically prone to the impacts of local-scale and severe weather (Ducrocq et al., 2014), which can dramatically affect the wellbeing and the economies of local communities (Rebora et al., 2013; Arrighi and Domeneghetti,2024).

The agreed protocol for regional climate projection delivery relies on the availability of standardized global climate projections from the international Coupled Model Intercomparison Project, now in its 6th phase (CMIP6, Eyring et al. 2016).

State of the art global projections typically have a nominal horizontal grid resolution of 100-200 km, corresponding to a

three/five-time larger effective resolution (Klaver et al., 2020). The necessity of better representing local processes and teleconnections among distant regions (Mahajan et al., 2018), as well as of directly providing boundary conditions to high-resolution Regional Climate Models (RCMs) with no need of an intermediate nesting (RCMs, Dickinson et al., 1989), has recently prompted very high-resolution (120÷20 km) coordinated global experiments in the CMIP framework (HighResMIP, Haarsma et al., 2016). These efforts have anyway proved to be too demanding in terms of computational and storage resources,

so that dynamical downscaling via RCMs still constitutes the most viable solution to describe the complex phenomena (Feser et al., 2011) and mesoscale interactions that emerge over complex morphology regions such as the Mediterranean (Doblas-Reyes et al. 2021).

As a matter of fact, IPCC AR6 acknowledged that regional climate projections now provide increasingly robust and mature information to feed climate services and impact studies at the necessary high resolution (Ranasinghe et al. 2021). RCMs are

similar to GCMs as to model architecture, but they are applied over limited areas and implemented as a boundary condition problem, with boundary information usually provided by a driving GCM. RCMs can both provide sub-continental climate information and improve process understanding. In analogy with the CMIP initiative, the COordinated Regional climate Downscaling Experiment (CORDEX) (Giorgi et al., 2009; Giorgi and Gutowski, 2015) provides a multi-model ensemble of present climate and future projections for different regions of the planet. Typical resolutions for the CORDEX models range

from 50 to 10 km. The CORDEX experiments cover 14 different regions, including the European region (EURO-CORDEX) and the Mediterranean region (MED-CORDEX, Ruti et al., 2016).

While the resolution of RCMs indeed improves the dynamical representation of atmospheric processes, sub-grid processes still need to be parameterized. In particular, convection by cumulus clouds exhibits a continuous energy spectrum across kilometer scales without apparent scale separation (Wyngaard, 2004; Moeng et al., 2010), prompting a sustained effort to further increase

resolution in both weather and climate simulations, up to an ideal limit that balances model accuracy and computational requirements (Prein et al., 2015). Convection plays a crucial role in vertically redistributing heat and moisture, thus modulating the vertical structure of the atmosphere and its stability and interacting with mesoscale dynamics. It triggers major impact drivers, such as heavy precipitation, windstorms and floods, whose representation is essential for environmental risk



assessments. Nevertheless, its parameterization constitutes a major source of uncertainty in model projections (Foley et al.,
2010, Ban et al., 2014) and bypassing it by explicitly representing all the involved spatial scales, from kilometers to (ideally)
tens of meters, is indeed an attractive solution. The added value of resolving the CP-scale has been demonstrated in terms of
both local circulation and land–atmosphere interaction description (Coppola et al., 2020, Soares et al., 2022, Sangelantoni et
al., 2023, Belusic-Vozila et al., 2023), as well as the capability of Convection Permitting Models (CPMs) to improve the
representation of precipitation extremes (Pichelli et al., 2021), allowing to investigate their sensitivity to global warming. In
multi-model studies, heavy precipitation events are in fact found to propagate farther and faster at the end of the century in a
RCP8.5 warming scenario, with an increase in precipitation volumes, hit area and severity (Muller et al, 2023, Caillaud et al.,
2024). In particular, Fosser et al. (2024) found that the CORDEX-FPSCONV CPM ensemble (Coppola et al., 2020) reduces
model uncertainties by more than 50% compared to lower resolution models, due to the more realistic representation of local
dynamical processes.

On the other hand, regional climate simulations typically span several decades and comparatively large domains and can still
prove very expensive in terms of computational resources (Fuhrer et al. 2018), limiting grid spacing within the bounds of the
so-called gray zone. While cloud-scale and updraft statistics only converge for scales of the order of tens of meters (Jeevanjee
2017; Panosetti et al. 2018, 2019) and storm morphology is still resolution-dependent below 1 km (Hanley et al. 2015), it is
generally assumed that deep convection at least is permitted for horizontal grid-spacing between 1 and 4 km (Weisman et al.
1997; Hohenegger et al. 2008, Kendon et al. 2017; Prein et al. 2015). The gray zone spans the 4÷10 km range, where the
performance of convection parameterization is critically scale dependent, scheme assumptions violations can potentially be
induced and the optimal resolution at which it is preferable to turn it off needs to be assessed (Vergara-Temprado et al., 2020).
As a matter of fact, high (6 km) and very high (2 km) resolution reanalysis over the European region have shown that a grid
spacing of 6 km is already sufficient to reproduce precipitation accumulation comparable to point observations, at least for
accumulation times larger than 1 hour (Wahl et al., 2017), although the higher resolution dataset improves point-to-point
comparison, due to the combined effects of data assimilation techniques, grid refinement and explicit convection. It is, in fact,
generally difficult to assess the relative weight of simultaneously implemented improvements in cross-resolution comparisons
(Vergara-Temprado et al., 2020).

In this context we present here results from an evaluation run (ERA5 driven) and from a coherent set of high-resolution multi-
scenario climate simulations (present climate, SSP1-2.6, SSP2-4.5, and SSP5-8.5, O'Neil et al., 2016 ) obtained by applying
a double-nesting technique to dynamically downscale the global CMIP6 MPI-ESM1-2-HR projections. The Weather Research
and Forecasting model (WRF) is used, first over the EURO-CORDEX domain, at a horizontal spatial resolution of 15 km, and
then over a finer grid covering the whole of Italy and extending to the north-western Mediterranean at a resolution of 5 km.
Future scenarios extend up to 2100.

The paper is organized as follows: section 2 presents the model characteristics and describes the protocol adopted, as well as
the independent datasets used for the evaluation; section 3 is dedicated to the evaluation analysis of the reanalysis driven
simulations; section 4 deals with the results of the scenario simulations; conclusions are summarized in section 5.



## 2 Model and data description

We produce a coherent set of high-resolution multi-scenario climate simulations based on the WRF-ARW version 4.2.2 (Skamarock et al. 2008). The model configuration was chosen in accordance with the guidelines provided by the WRF modelling community for the coordinated runs in the context of the EURO-CORDEX CMIP6 protocol. The WRF community adopted these guidelines to perform evaluation and scenario simulations driven by different Global Climate Models, and to facilitate the exchange and use of data, also as boundary conditions for higher resolution models.

We use a double-nested domain strategy to downscale the coarse global CMIP6 data from a regional domain, covering the whole Europe, to a fine spatial scale domain centered over Italy (as represented in Figure 1 trough orography).

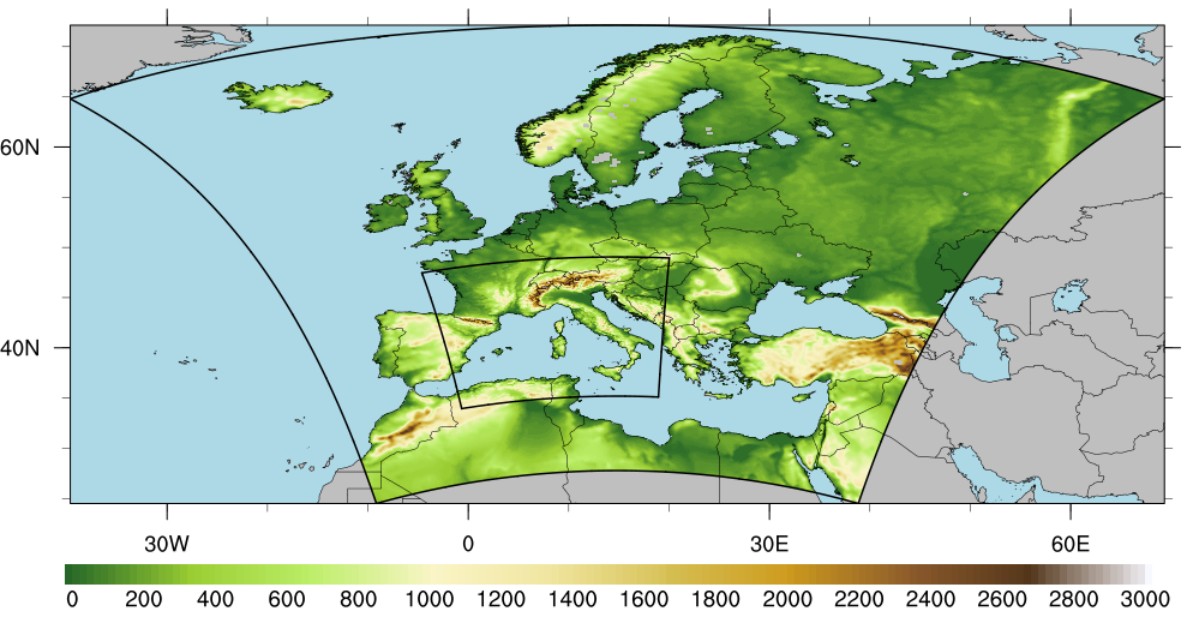

**Figure 1: Model domains used for the downscaling with WRF regional model: European domain (D01- 15km), national domain (D02- 5 km). Colors represent the orography of the region.**


The first domain (D01) has a horizontal grid resolution of 15 km, while the nested domain (D02) has a resolution of 5 km. Both domains use a Lambert conformal projection. The resolution of the inner domain falls in the so-called gray zone for the representation of deep convection: this choice enables us to cover relatively wide-size domains of regional interest and still allows for transient simulations, whether historical or projections, that are long enough to give some robustness to the statistics





of extreme events. In any case, one should be cautious in order not to run into ambiguous results due to the use of parameterizations that are not suitable for the simulation resolution. In this context, the Weather Research and Forecasting model with the Advanced Research core (WRF-ARW, Skamarock et al., 2008) provides a wide suite of parameterizations, including scale-aware ones that have proved effective in managing the transition to more resolved scale (Liu et al., 2011, Jeworreck et al., 2019, Park et al., 2024). Table 1 summarizes the configuration implemented for the experiments described in

this work.

For the representation of deep convection, we have implemented the cumulus parameterization proposed by Grell-Freitas (Grell and Freitas, 2014; Freitas et al, 2021) (see Table 1).

| Parameterization | Type | Short description | Reference |
|---|---|---|---|
| Cumulous | Grell-Freitas | Convective cloud representation | Freitas et al, 2021 |
| Microphysics | Thompson | Simulates the transport, physical change, and thermodynamic effects of the total hydrometeor population in clouds | Thompson et al., 2008 |
| PBL | MYNN 2.5 | Turbulence representation in the lower troposphere | Nakanishi and Niino, 2009 |
| Land Surface | Noah_MP | Simulate the exchange of water and energy fluxes at the Earth surface–atmosphere interface | Niu et al., 2011; Yang et al., 2011 |
| Radiation | RRTMG | The Rapid Radiative Transfer Model for GCMs for interaction with solar radiation at SW and LW spectral-band | Iacono et al., 2008 |

**Table 1. Main parameterizations adopted for the downscaling with WRF regional model**


Although there is not a universal recipe about the sub-grid physics configuration, and the choice may strongly depend on the specific case and region of interest (for a review see, for example, the introduction of Jeworrek et al., 2019), there is consensus in the adoption of a scale-aware parameterization of convection such the Grell-Freitas one (Grell and Freitas, 2014; Freitas et al, 2021, Jeworrek et al.,2019, Park et al.,2024) for experiments dedicated to investigate the benefits of increasing the grid

resolution.

To assess the performances of the model in both the regional and fine-spatial scale domains, a hindcast simulation forced by ERA5 reanalysis (Hersbach et al. 2020) has been produced for the period 1980-2023.



Historical and three future scenario simulations (SSP1-2.6, SSP2-4.5, SSP5-8.5, Eyring et al. 2016; O'Neill et al. 2016) have been produced downscaling the CMIP6 MPI-ESM1-2-HR (Gutjahr et al. 2019), which has a grid of T127 (0.93° or ~ 103 km).

Among all the available CMIP6 models, in addition to the relatively high spatial resolution, we selected the MPI-ESM1.2-HR as it has a well-balanced radiation budget and its climate sensitivity is explicitly tuned to 3 K (Müller et al. 2018), making this model well suited for prediction and impact studies.

The present-climate experiment (historical) covers a period of 35 years, nominally from 1980 to 2014 (overlapping great part of the hindcast simulation), while future climate simulations span the period 2015-2100.

Table 2 synthetically shows the newly produced simulations, providing some details. For each simulation an identification of the experiment has been assigned and domain, resolution and length are listed.

| ID | Domain ID | H Res [Km] | Start | End | N years | Forcing | Simulation |
|---|---|---|---|---|---|---|---|
| 15Km-Hindcast | D01 | 15 | 1980 | 2023 | 44 | ERA5 | hindcast |
| 15Km-Historical | D01 | 15 | 1980 | 2014 | 35 | MPI-ESM | historical |
| 15Km-SSP126 | D01 | 15 | 2015 | 2100 | 86 | MPI-ESM | SSP126 |
| 15Km-SSP245 | D01 | 15 | 2015 | 2100 | 86 | MPI-ESM | SSP245 |
| 15Km-SSP585 | D01 | 15 | 2015 | 2100 | 86 | MPI-ESM | SSP585 |
| 5Km-Hindcast | D02 | 5 | 1980 | 2023 | 44 | ERA5 | hindcast |
| 5Km-Historical | D02 | 5 | 1980 | 2014 | 35 | MPI-ESM | historical |
| 5Km-SSP126 | D02 | 5 | 2015 | 2100 | 86 | MPI-ESM | SSP126 |
| 5Km-SSP245 | D02 | 5 | 2015 | 2100 | 86 | MPI-ESM | SSP245 |
| 5Km-SSP585 | D02 | 5 | 2015 | 2100 | 86 | MPI-ESM | SSP585 |

**Table 2: Overview of the newly produced CMIP6 simulations with the RCM, according to the adopted protocol**

For the evaluation we use the following benchmarks: E-OBS (Dickinson et al. 2018), which is a daily gridded land-only observational dataset over Europe at 11 km resolution; the hindcast driving ERA5 reanalysis data (25 km) and two other reanalysis products, namely, ERA5-land (11 km, Muñoz-Sabater et al., 2021) and CERRA (5 km, Ridal et al., 2024). The latter is comparable, for its resolution and use of convection parameterization, to our highest resolution simulation (D02).





## 3 Evaluation of RCMs simulations

In the following sub-sections, we evaluate the hindcast simulations analyzing both climatology and interannual variability of the near surface temperature (T2m) and of the total daily precipitation (P) and comparing the results with available observational datasets and/or independent reanalysis datasets. These Essential Climate Variables (ECVs) are chosen because they affect a wide variety of processes with important implications for natural ecosystems and human society.

Although the hindcast simulations are available for the period 1980-2023, we perform the evaluation over the common period
among the different datasets used as benchmarks, i.e. 1984-2014.

### 3.1 Climatology and interannual variability

In order to quantify the climatological biases of the 15km-Hindcast experiment across its entire domain, we computed the mean seasonal cycles over the PRUDENCE European geographical subregions (Christensen and Christensen 2007), which are commonly used as standard regions for the evaluation of the EURO-CORDEX climate simulations ( e.g. Kotlarski et al. 2014):
British Islands (BI), Iberian Peninsula (IP), France (FR), Middle Europe (ME), Scandinavia (SC), Alps (AL), Mediterranean (MD), Eastern Europe (EA). The D01 domain covers all these regions, while only the Alps domain entirely falls within the D02 area. Figures 2 and 3 show, respectively, the seasonal cycles of temperature and precipitation on each PRUDENCE sub-domain for the 15km-Hindcast simulation. The seasonal cycle derived from the ERA5 dataset (driver) is also shown for comparison. Results indicate that the 15km-Hindcast simulation closely follows the driver's seasonal mean curve of T2m and
P. In particular, the temperature bias of the downscaled model is within 0.5-1.5 °C over all the subdomains considered on an annual basis, with more evident deviations in winter months. Moreover, the model is fairly performing in the Mediterranean domain. Precipitation seasonal cycles show a systematic wet bias with respect to the driver, across the whole year over some sub-regions, like France and Alps, while in the other sub-regions (ME and EA) the bias peaks during the warm season. However, such biases are mostly within 1 mm/day, only exceeded in the JJA season in the Eastern Europe subdomain.




**Figure 2: Seasonal cycle of the T2m(°C) on the Prudence subdomains: ERA5 reanalyses (blue), 15-Hindcast simulation (red).**







**Figure 3: Seasonal cycle of the total precipitation P(mm/day) on the Prudence subdomains: driving ERA5 reanalysis (blue), hindcast simulation 15Km-Hindcast (red).**




To investigate the effect of increasing the resolution, we compute the mean seasonal bias, i.e. the differences of the seasonal means of the T2m and P against E-OBS, for the two simulation domains.

All the fields have been re-gridded to the coarsest resolution of 15km. Figure 4 shows in the first column the seasonal means of T2m and P, derived from the reference observational dataset E-OBS. Data are shown only over land due to the E-OBS

availability and over the common area of D02 domain. The second and third columns represent the model bias of the seasonal average against the observations for the 15km-Hindcast and 5km-Hindcast simulations, respectively.

There is a prevailing cold bias in each season within 2°C, although greater values are reached in mountainous regions. Conversely a warm bias is evident over plain regions, especially in the Po Valley (southern flank of the Alps in northern Italy). Increasing the resolution, we observe a slight reduction of the temperature bias.

The improvement is much more evident for the precipitation field, especially in summer season, when the effects of convective precipitation are expected to be more relevant; the positive bias of 15km-Hindcast with respect to observation is clearly reduced in 5km-Hindcast, showing the added value of the double nesting procedure increased resolution. A slight overestimation of P is still evident over the orography across the Alpine region.







**Figure 4: Seasonal model bias (second and third columns) with respect to E-obs (first column) of T and P for the D01 and D02 hindcast experiments. The row panels represent DJF, MAM, JJA, and SON seasons.**



Figure 5 shows the precipitation seasonal cycle averaged over the AL sub-region, for both the downscaling domains against ERA5, E-OBS and CERRA dataset. Sea points falling in the domain have been masked out. The three-reference datasets are quite close to each other over the whole annual cycle, falling within a range that does not exceed 1 mm/day. The 15km-Hindcast experiment (red) has a wet bias with respect to all the reference datasets throughout the year of the order of 1 mm/day and 2 mm/day compared to E-OBS and ERA5, respectively. The 5km-Hindcast curve (orange) reproduces a more realistic

seasonal variability, similar to the observed one, which is characterized by the two relative maxima during spring and autumn, clearly reducing the biases compared to its 15km-Hindcast driver within April and November.

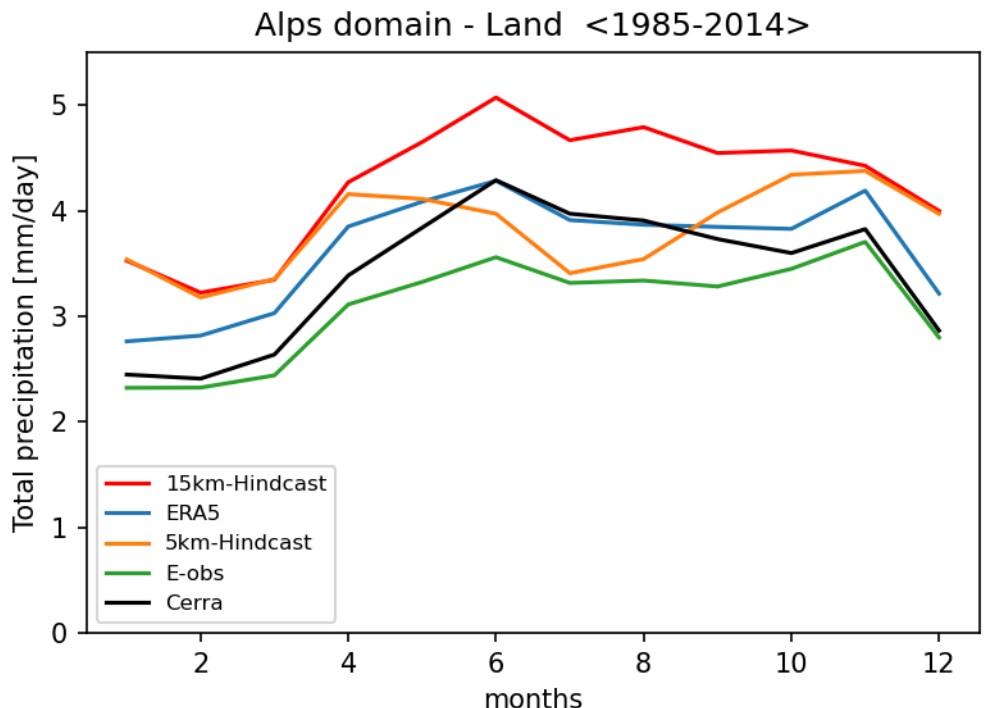

**Figure 5: Seasonal variability of P over the Alps domain (AL) averaged over land only. Experiments 15km-Hindcast (red), 5km-Hindcast (orange). Reference datasets: ERA5 (blue), E-OBS (green), CERRA (black)**

In Figure 6 we analyzed in detail how the two contributions to the total daily precipitation are produced by the schemes of cumulus and microphysics in the hindcast simulations. The seasonal cycles of the stratiform precipitation (long dash) and convective precipitation (points) in the 15km-Hindcast (red) have maxima of the same order of magnitude, although in different seasons, as expected. By comparison with the 5km-Hindcast curves (orange) and keeping in mind the results in Figure 5, we can speculate that an overestimation of the contribution coming from the cumulus parameterization during summer is the cause

for the wet bias in the 15km-Hindcast. On the other hand, the contribution of the convective parametrized precipitation in the 5km-Hindcast (orange points) is one order of magnitude lower than the one at coarse resolution for every month of the year, thus most of the precipitation is managed by the microphysics parameterization. This suggests that the model in the gray-zone,





thanks to the scale-aware behavior implemented in the Grell-Freitas scheme (Freitas et al, 2021), mimics a convection permitting model, smoothing the transition toward the km scale.

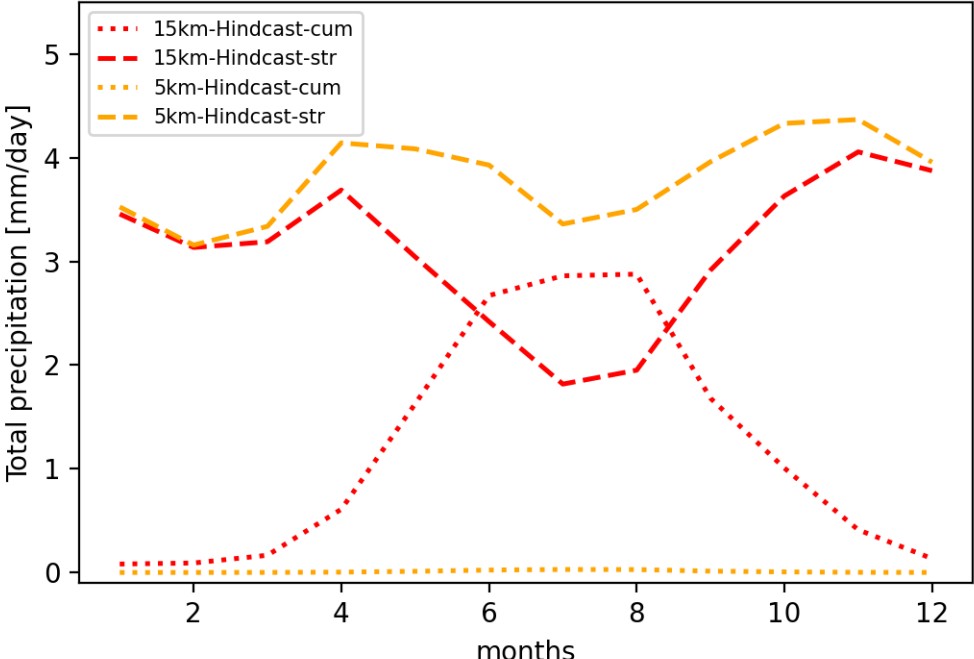

**Figure 6 Partitioning of the total precipitation among cumulus (cum) and stratiform (str) ones for the two hindcast experiments**

Figure 7 shows the interannual variability of the daily precipitation over the Alps domain. The other PRUDENCE domains have also been analyzed for the 15km-Hindcast simulation but are not shown for brevity reasons. It is worth noting anyway that the 15km-Hindcast simulation closely follows its driver in terms of T2m with biases within 0.5°C, more evident over the IP (overestimation) and the SC, MD and EA sub-regions (underestimation). Also in terms of precipitation the 15km-Hindcast closely follows ERA5, with only a slight overestimation tendency within the 1 mm/day for most of the sub-regions. The highest wet bias is reached in the Alps region (Figure 7); however, the bias doesn't exceed 1 mm/day along the whole time series with respect to its driver and 2 mm/day with respect to the E-OBS dataset. The model bias is considerably reduced in the 5km-Hindcast experiment, and mainly attributable to the bias reduction over the spring-to-fall period discussed for the annual cycle (Fig. 5).





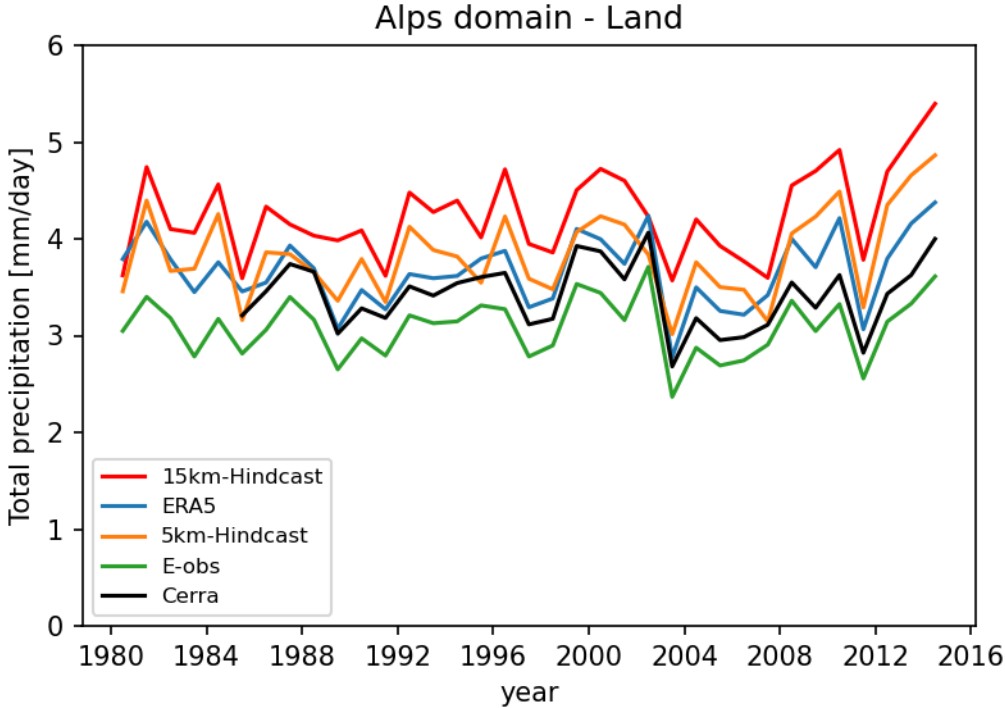

**Figure 7 Interannual variability of P over the Alps domain (AL) averaged over land only. Experiments 15km-Hindcast (red), 5km-Hindcast (orange). Reference datasets: ERA5 (blue), E-OBS (green), CERRA (black)**

## 3.2 Statistics of extreme events

The Mediterranean is a region particularly prone to heavy precipitations, due to the complexity of its morphology which reflects in the interactions between local and large scale forcings which trigger them. For example, its orography is a key factor for flow modifications that drive intensity, location and duration of orographic rain (Rotunno and Houze, 2007); the Alps' shape favours lee-cyclogenesis (Buzzi et al., 2020), which can trigger heavy precipitations over sub-regions lying at their Southern flank (Rotunno and Ferretti, 2001). Moreover, the sea, as source of moisture, can modulate the intensity of the precipitation, reinforcing the vapour load of low-level jets converging over orography (Buzzi et al., 1998). A fair representation of heavy precipitation is crucial to study their sensitivity to the global warming. The following analysis evaluates the model ability in reproducing the most intense rainfalls over Italy.

Figure 8 compares the Probability Density Functions (PDFs) of daily precipitation over Italy for different datasets: ERA5, CERRA, E-OBS, and the two hindcast experiments. The statistics have been computed over the common period 1984-2014. Data are reported on their original grid, and each event is defined as the daily precipitation at each grid point. The distributions are normalized with the dataset total number of events. For a fair comparison, especially in terms of extreme events, a model simulation should be compared with a benchmark with the closest possible resolution. As already discussed in previous





sections, the 15km-Hindcast simulation (red line) overestimates the observations (green line) across the whole PDF and
especially at its tail, producing quite large extremes for such a resolution (400 mm/day), closer to the CERRA dataset which
has a much higher resolution. The comparison between the 5km-Hindcast and the CERRA PDFs shows that, although the
experiment still overestimates its reference, it slightly reduces their distance in terms of extremes, catching the rarest ones at
the tail of the distribution, but overestimating their precipitation. Although less clearly in terms of extremes, these results are
in line with the indication of an added value of the increased resolution for simulating the precipitation.

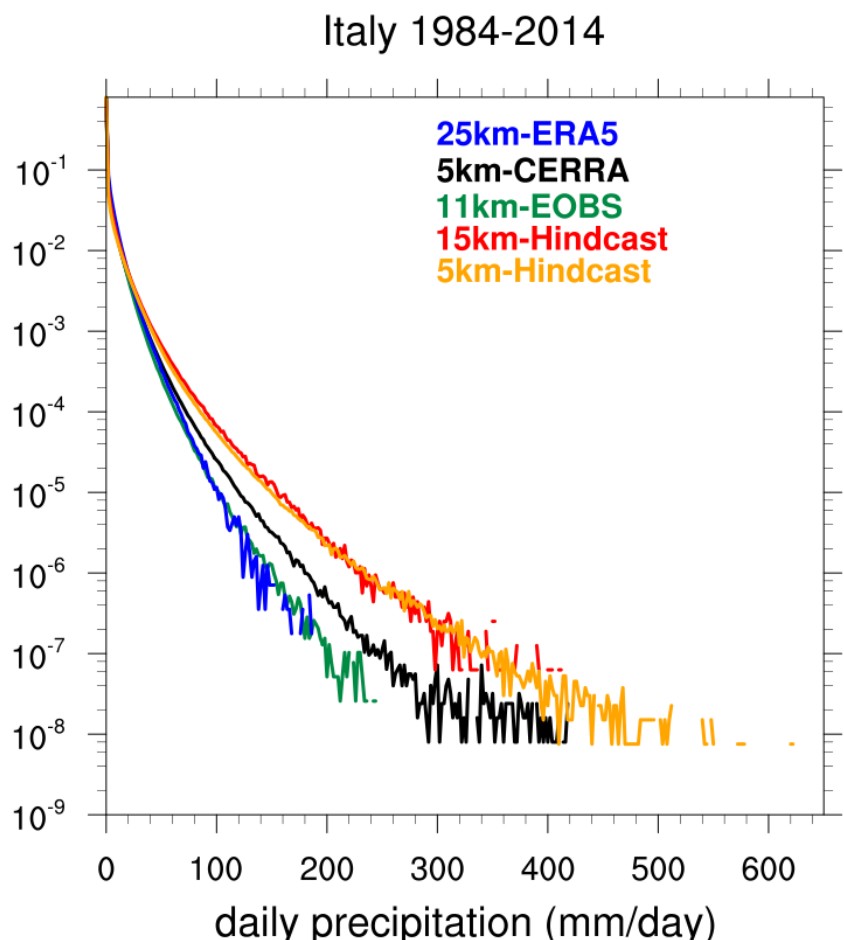

**Figure 8: Probability density function of the daily precipitation over Italy: ERA5 (blue), CERRA (black), E-OBS (green) and the experiments 15km-Hindcast (red) and 5km-Hindcast (orange) over the period 1984-2014.**

Figure 9 shows more in detail the geographical distribution of the average heavy precipitation represented through the 95th, 97th and 99th percentile of the two experiments 15km-Hindcast and 5km-Hindcast and of reference datasets.



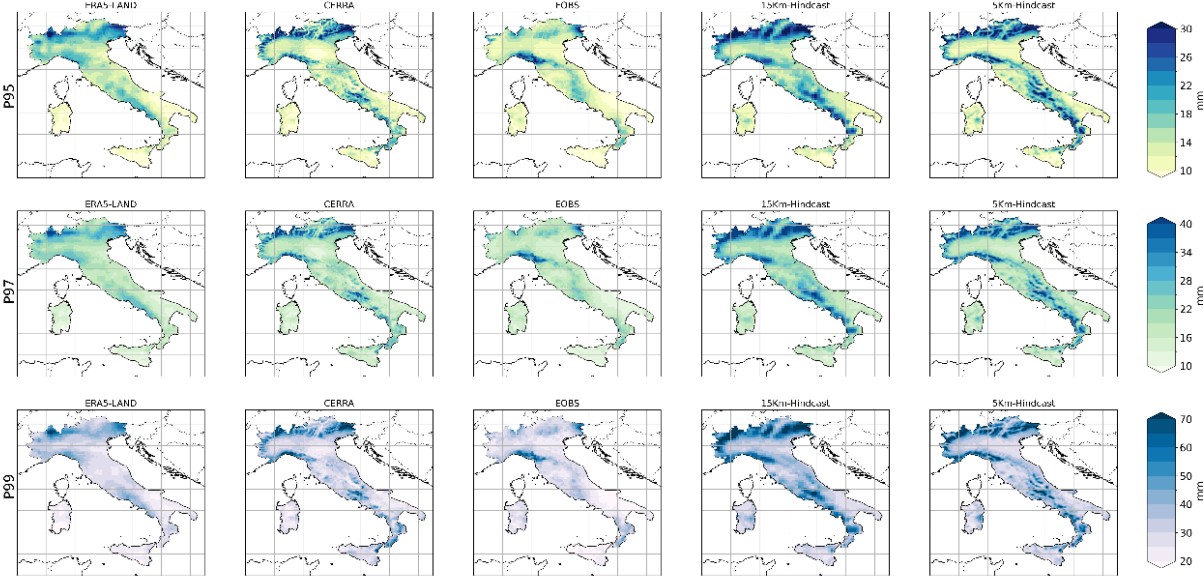

**Figure 9 Annual P95 (top), P97 (mid), and P99 (bottom) of daily rainfall from ERA5-LAND, CERRA, E-OBS and the simulations 15Km-Hindcast and 5Km-Hindcast.**

The most severe events are concentrated in the mountain regions: in the alpine sector reanalysis and experiments give coherent
results and overestimate the observations, although the two experiments show some differences. It is worth noting that part of the model bias over orography compared to E-OBS might not be accounted as an error due to possible under-representation of heavy precipitation within the observational dataset due to poor spatial coverage of rain gauges feeding the data and to possible under-catchment issues (La Barbera et al., 2002). In the Ligurian region and north Apennines CERRA, E-OBS and the two experiments have comparable results with respect to spatial distribution and intensity; in the central and meridional regions
the overestimation with respect to the reference datasets is more evident at all the considered percentiles.

## 4    Results from the scenario simulations

The simulation protocol includes a historical simulation for each domain. The two experiments 15km-Historical and 5km-Historical, which must be used as a term of reference for future impact assessment, were run with the same configuration as the hindcast one but were forced by the CMIP6 MPI-ESM1-2-HR model. The historical simulation aims to reproduce the main
statistics of the current climate. The climate change signal is computed as the average difference between 2071-2100 and 1985-2014 periods. Figures 10 and 11 show the spread of T2m and P derived from the 15km-Historical experiment over the PRUDENCE domains, compared to the driving GCM and to the ERA5 as a term of reference for current climate.



**Figure 10: Box plot of historical simulation of T2m: mean (yellow points), median (orange lines within the box), 1st-3rd interquartile range (IQR, color box). The whiskers extend from the box to the farthest data point lying within 1.5x the inter-quartile range (IQR) from the box. 15km-Historical experiment (red), ERA5 (black), GCM (green).**



**Figure 11: Same as Figure 10 but for mean precipitation.**


The historical experiment does not show large differences with respect to the results previously reported in section 3 for the hindcast: the climatological mean of the surface temperature averaged over the domains is generally close to the driver and within its variability, although in some areas it is closer to ERA5 than to the driver (IP, FR). The climatological values are also close to those derived from ERA5 and can be fairly considered representative of current climate temperatures. In terms of

precipitation, the global driver is close to the ERA5 dataset, with no-definite bias tendency across the domains. The bias is positive in some regions and negative in others but always within 1 mm/day. The downscaled precipitation field suffers from the same characteristics already analyzed in the hindcast experiments. There is a wet bias in all regions (excepting SC domain) that is corrected, analogously to what shown for hindcast simulation in Fig.5, with the double-step nesting at 5km in the Alps domain (not shown).





Figure 12 shows the 2m-temperature projected climate change in the three scenarios over D01 domain for the four seasons. A general warming is shown, as expected, for each scenario on the whole region. This is in accordance with analogous regional experiments driven by the same global driver but conducted within the Med-CORDEX protocol with the coupled model ENEA-Reg (Anav et al., 2024), adopting the same version of the atmospheric model but with different parameterizations. Scenario SSP1-2.6 does not display large changes among the seasons while in SSP2-4.5 and SSP5-8.5 the change signal evidences seasonal differences. In particular, the largest changes are projected in DJF over East Europe and JJA in the Mediterranean. Values at all grid points are significant at 10% level. The significance has been assessed by a Monte Carlo bootstrap procedure with 1000 repetitions.

**Figure 12: Projections of temperature change at the end of the century in the multi-scenario experiments – 15km SSPs**

Figure 13 shows the projections of precipitation change at the end of the century. Black dots indicate 10% level significance, assessed by Monte Carlo bootstrap procedure with 1000 repetitions. Even in the mitigated scenario SSP1-2.6, significant





changes are present. For all the scenarios and seasons there is a projected intensification of the hydrological cycle over most of the continental EU and precipitation reduction over the Mediterranean region. In JJA the precipitation reduction extends also to West EU (Spain and France). Similar results were found in Anav et al. 2024, with local differences likely due to the

different configuration of the model (ocean coupling and parameterizations).



**Figure 13: Projections of precipitation change at the end of the century in the multi-scenario experiments 15km-SSPs**

Figures 14 and 15 are analogous to figures 12 and 13 but for the D02 domain. The warming over the national territory in 5km-

SSP1-2.6 (Fig 14, left column) is mostly contained within the range 0.5 – 1 °C across the different seasons. During fall the projected climate change reaches 1.5 °C over Italy and the Tyrrhenian sea. The 5km-SSPs experiments tend to maintain a mean projection of the future warming like that of the coarser domain, although the effect of increased resolution reduces the





warming in JJA and SON over most of the peninsula and especially in DJF and MAM in the Po valley, as can be seen by comparing figures 12 and 14.


**Figure 14 Projections of temperature change at the end of the century in the multi-scenario experiments – 5km SSPs**

Figure 15 shows the projected climate change in precipitation for the three scenarios, across the seasons. In the 5km-SSP1-2.6 experiment the mean change in the precipitation field over Italy at the end of the century in DJF and MAM is generally positive and exceeds 0.5 mm/day in mountain regions, especially in the alpine sector. This result is in accordance with the results of

the 15km-SSP1-2.6 as to the sign and strength of the signal, but more statistically robust.

In JJA season the 5km-SSP1-2.6 experiment projects a significant reduction in the precipitation field almost over the entire Italian territory. Instead, in the 15km-SSP1-2.6 there is not a clear precipitation change and Italy stands in the transition between two distinct zones with opposite change, i.e. Western and Eastern Europe.

For the SON season no large differences are evident in the projections between the two different resolutions.




**Figure 15: Projections of precipitation change at the end of the century in the multi-scenario experiments 5km-SSPs**

The projected change of the mean precipitation for the scenarios SSP2-4.5 and SSP5-8.5 are similar across the seasons. The
climate becomes generally drier especially in summer, while in the alpine region the mean precipitation is expected to increase
in winter and fall, thus projecting an increase of the seasonality of the hydrological cycle in this area.

Climate change is expected to alter the frequency and intensity of extreme precipitation events, which initiate natural hazards
such as floods, or may trigger landslides. Extreme convective precipitation events are getting more intense and more frequent
due to global warming, hitting larger areas and exacerbating their characteristics, especially over the Mediterranean (Pichelli
et al., 2021, Muller et al., 2023, Pichelli et al., 2023, Caillaud et al., 2024). Figure 16 shows the change of the precipitation
PDFs at the end of century for the three scenarios. The PDFs computed at the end of the century (2071-2100) are shown in
bright colors, while the PDFs computed over the reference period (1984-2014) are shown with the same fading color. In the



sustainable scenario (SSP1-2.6) we do not detect significant variations among present and far future projections globally over Italy. However, the SSP2-4.5 and the SSP5-8.5 scenarios project an increase in both frequency and intensity of the extreme

events, which are similar between the two resolution runs within the range of 150-300 mm/day thresholds, while it is exacerbated, especially in terms of frequency, at 5km respect with its 15km driver at the very end of tail of the distribution. More detailed information about the changes of the extreme events in the far future can be derived from figures 17 and 18.

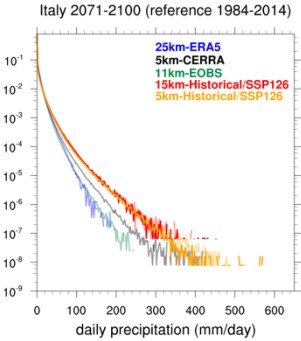
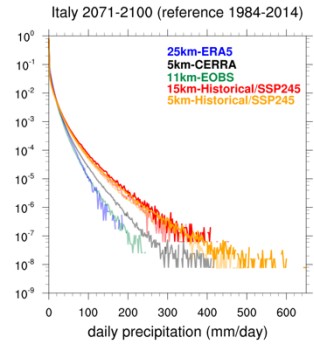
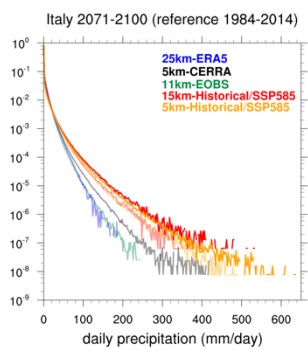


**Figure 16: Intercomparison of the precipitation distributions at the end of the century (2071-2100 / solid line) with the reference distributions (1985-2014 / faint line) for the three scenarios: SSP1-2.6, SSP2-4.5 and SSP5-8.5 from left to right.**

Figures 17 and 18 show the change at the end of the century of the 99th percentile (P99) of daily precipitation over the Italian

territory for the different seasons and for the two experiments, respectively. By using a Jackknife method to calculate the P99 values, we reduce the bias of the estimated P99 values (Ferreira, 2023). At the same time, the Jackknife methodology provides an estimate of the mean squared error of the P99 values for the scenario and for the reference period, which allows us to perform a t-Student test at the 5% confidence level that the modelled changes in the intensity of extreme events are statistically significant. The Jackknife method is applied by dropping half of the original samples before calculating the P99 values and

the corresponding difference between the scenario and the reference period. Except for a few grid points corresponding to areas where the difference in P99 is very small, most of the differences shown in Figures 17 and 18 are statistically significant. There is a general agreement in the results of the two experiments, with limited local differences and with an overall reduction in the amplitude of the climate change signal in the higher resolution experiment.

It is worth noting that it is quite difficult to identify a general consistent trend across the climate scenarios from SSP126 to

SSP585, with the exception of SON season. As an example, the intensity of extreme precipitation along the Tyrrhenian coast of southern Italy increases during DJF in SSP126, while the scenario SSP245 shows a decrease in intensity. On the contrary,



SSP585 is similar to SSP126 in this area. During DJF and MAM, the scenarios from SSP126 to SSP585 show a gradual reduction in the intensity of the extremes over the northern part of the Apennines, along the northern coast of the Adriatic Sea. However, a well-defined seasonal pattern of changes in extreme precipitation can be observed. During DJF, there is an increase in intensity in the western Alpine region and a slight decrease in intensity in the eastern Alps, while the intensity of extreme precipitation decreases in the south, with a particularly marked decrease over the most prominent reliefs in Sicily; a similar pattern occurs during MAM, except for an increase in the intensity of extremes over the entire Alpine region; during JJA, there is a decrease in the intensity of extremes, with a more pronounced signal in the 5 km experiment and over the western coast of the peninsula, along the Tyrrhenian Sea. During SON an increase in the intensity of extreme precipitation is particularly relevant for the SSP5-8.5 for most of the area analyzed and for both the D01 and D02 experiments, where an evident increase of intensity of the p99 events is detectable in correspondence of the increase of scenario severity.





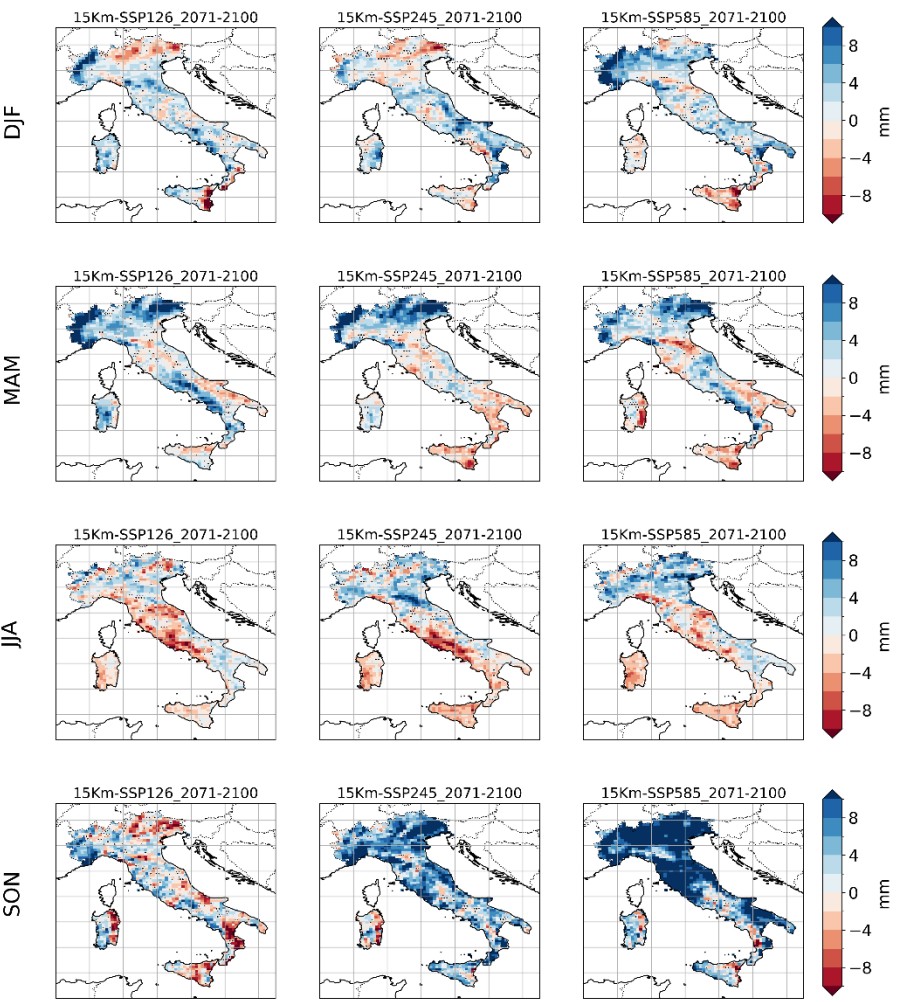

**Figure 17 Difference between the values of the 99$^{th}$ percentile (P99) of daily rainfall during the period 2071-2100 and the corresponding values computed for the period 1985-2014 The differences are computed for the three scenarios for the model configuration D01. Dotted areas correspond to the grid points where the difference is not statistically significant according to a t-Student test at 5% confidence level applied to a synthetic population of P99 values generated with the jackknife method described in section4**



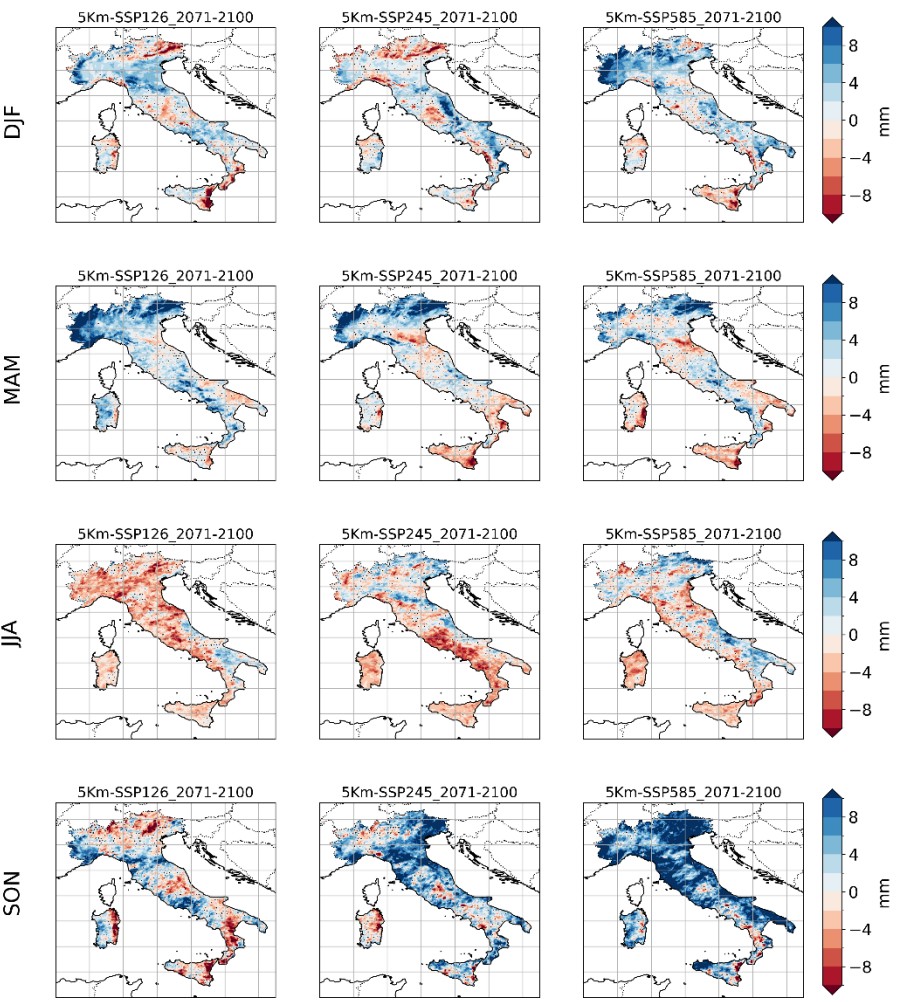

**Figure 18 Same as Figure 16 but for the model configuration D02**

## 5    Conclusions

A dynamical downscaling strategy has been developed and validated to produce multi-scenario regional climate simulations with the atmospheric model WRF for the Mediterranean region; the finest level of nesting (5km) is focused over the Italian Peninsula and the Western Mediterranean basin, while its intermediate parent domain has a resolution of 15 km over Europe.



The evaluation of the protocol simulation has been carried out through an ERA5 driven experiment. The realization of hindcast runs is of primary importance as it allows to test the ability of the numerical tool to reproduce the current climate and to validate the system against the reanalysis and observational datasets.

In this work, we explore a grid-step (5km) within the range of the so-called gray-zone (4-10km), usually avoided because at these scales the convection parameterizations should be still turned on being the convection still insufficiently resolved with

the risk of some parameterization assumptions being potentially violated. We proved that the model in the finer gray-zone, due to the scale awareness implemented in the Grell-Freitas cumulus scheme, mimics a convection permitting model behavior smoothing the transition toward the km scale.

Doing so, we are able to cover wide enough domains but saving both computational time and storage, and to perform scenario simulations long enough to speculate on the impacts of extreme events under climate change conditions.

Especially in terms of representation of mean precipitation, the 5-km simulations are proved to correct a wet bias present in the intermediate 15-km simulations over the most part of the Italian peninsula, revealing the added value of the double nesting procedure. This feature is confirmed also by analyzing the statistics of extreme rainfall events against reference datasets.

The production of a coherent set of high-resolution multi-scenario (SSP1-2.6 , SSP2-4.5 and SSP5-8.5) climate simulations of the last generation of global models (CMIP6) for Italy is by itself a novelty. Although several products are already available

at comparable or higher resolutions, they are in some cases limited to current climate as they are reanalysis or hindcast products (Giordani et al.,2023; Viterbo et al., 2024), while scenario projections are still referred to CMIP5 drivers and are available either at mid-century or on short time slices (Pichelli et al. 2021; Raffa et al. 2023).

For all the scenarios and seasons there is a projected general warming along with an intensification of the hydrological cycle over most of the continental EU and mean precipitation reduction over the Mediterranean region accompanied, over the Italian

Peninsula, by a strong increase in the intensity of extreme precipitation events, particularly relevant for the SSP5-8.5 scenario during autumn.

Finally, let us remark that both the intermediate and high-resolution simulations are adequate to provide the boundary conditions for convection-permitting scale (finer than 4 km) further downscaling. This will be the subject of future work, along with the investigation of any possible improvements deriving from increasing the resolution, especially in relation to known

issues about the convection representation (early on-set, drizzle problem, under-representation of extreme intensity precipitation).

Very high-resolution simulations may be planned over limited areas following the needs of impact researchers. Further room for improvement is expected by moving to an explicit simulation of convection and the comparison with the results obtained in the gray-zone will be of particular interest in the perspective of using climate data to produce relevant information, in terms

of climate indicators for climate services.





**Code and data availability**

The current version of WRF is available from the project website https://github.com/wrf-model/WRF/tree/v4.2.2 (WRF-ARW DOI doi:10.5065/D6MK6B4K , last accessed 31/01/2025) under the licence:

*WRF was developed at the National Center for Atmospheric Research (NCAR) which is operated by the University Corporation for Atmospheric Research (UCAR). NCAR and UCAR make no proprietary claims, either statutory or otherwise, to this version and release of WRF and consider WRF to be in the public domain for use by any person or entity for any purpose without any fee or charge. UCAR requests that any WRF user include this notice on any partial or full copies of WRF. WRF is provided on an "AS IS" basis and any warranties, either express or implied, including but not limited to implied*

*warranties of non-infringement, originality, merchantability and fitness for a particular purpose, are disclaimed. In no event shall UCAR be liable for any damages, whatsoever, whether direct, indirect, consequential or special, that arise out of or in connection with the access, use or performance of WRF, including infringement actions.*

The ERA5 dataset is freely accessible after registration from the Copernicus Climate Data Store at https://cds.climate.copernicus.eu/datasets. Because of the large volume (>300 TB), the WRF output data and the scripts used

for the visualization in this paper can only be made available upon request.

**Author contribution**

GP, AD and AA designed the experiments, and AA carried them out and performed the simulations. AA, MA, EP, SC, and FC made data curation and formal analysis. MVS conceptualized the work and prepared the manuscript with contributions

from all co-authors.

**Competing interests**

The authors declare that they have no conflict of interest.

**Acknowledgements**

We acknowledge the World Climate Research Programme, which, through its Working Group on Coupled Modelling, coordinated and promoted CMIP6. Within this we thank the CMIP6 endorsement of the High-Resolution Model Intercomparison Project (HighResMIP) and Martin Schupfner for providing additional data from the MPI-ESM. The computing resources and the related technical support used for this work have been provided by CRESCO/ENEA-GRID High Performance Computing infrastructure and its staff.

**Financial support**

This study was carried out within:



Project KNOWING that received funding from the European Union's Horizon Europe research and innovation programme under grant agreement No 101056841" funded by the European Union – GA Project 1011056841.

RETURN Extended Partnership  that received funding from the European Union Next-GenerationEU (National Recovery and
Resilience Plan – NRRP, Mission 4, Component 2, Investment 1.3 – D.D. 1243 2/8/2022, PE0000005)

ICSC Italian Research Center on High-Performance Computing, Big Data and Quantum Computing  that received funding from the European Union Next-GenerationEU (National Recovery and Resilience Plan – NRRP, Mission 4, Component 2, Investment 1.4 – D.D: 3138 16/12/2021, CN00000013)

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
