# Peer review of "Impact of spatial resolution on multi-scenario WRF-ARW simulations driven by the CMIP6 MPI-ESM1-2-HR global model: a focus on precipitation distribution over Italy"

_EGUsphere, 2025_

## Author Comment (AC2)

The authors run two WRF models, a 5-km centered on Italy and a 15-km centered on Europe. The nested models are forced with ERA5 boundary conditions for an historical run and with 3 SSP scenarios using the MIP-ESM model.

> **REPLY:**
> *Thank you for your valuable feedback on our manuscript, which gives us the opportunity to significantly improve our work and clarify a few critical points.*
> *In fact, we are under the impression that some comments, and specifically those questioning the relevance of our paper to the journal's scope and focus and the overall consistency of our approach, might stem from an over-compressed description of the simulation protocol, which can lead to misunderstanding the hierarchical steps of the adopted double-nesting procedure.*
>
> *Similarly, we are concerned that the main point of the paper as stated in the abstract and the introduction, i.e. the analysis of a climate simulation run at a grid-step that falls within the so called gray zone for the convection representation (4-10 km), as well as its bearings on the optimal design of numerical simulations, was not fully appreciated. Nevertheless, we acknowledge that our line of reasoning could have been better highlighted along the article, and will proceed to amend the text accordingly.*

They describe this a CMIP6-driven, but really it is only a single model. For that title I would expect multiple CMIP6 models to be used. Please remove CMIP6 from the title.

> **1.REPLY:**
> *We acknowledge that the current title might suggest expectations greater than those we are dealing with. However, the results shown in this paper are among the very first obtained when downscaling a CMIP6 global model and this is a distinctive feature of our work. We propose to change the title with "Impact of spatial resolution on multi-scenario WRF-ARW simulations driven by the CMIP6 MPI-ESM1-2-HR global model: a focus on precipitation distribution over Italy".*

Basically, this is a paper not about model development, but about the simple application of an off-the-shelf WRF model run at two resolutions.

> *2. REPLY:*
> *We trust that once clarified what double-nesting is, some of the subsequent criticisms of the reviewer will be answered as well, and the main results of the paper might be fully understood.*
> *In particular, we will expand section 2 in order to include a more thorough description of the numerical protocol.*
> *In order to ease the current discussion, we anticipate here a brief illustration of how this protocol generally works:*
> 1. *a parent simulation at a relatively coarse resolution (i.e. 15km) is run as a first regional downscaling on a domain much larger than the final domain of interest, whose boundary and initial conditions (BC and IC) are provided by a global model (reanalysis or climate scenario simulations),*
> 2. *a second child simulation (i.e. 5km) is nested into the first, taking its boundary conditions from the parent simulation (i.e. 15km) and covering the inner domain of interest at higher resolution.*
> *This approach allows to perform a progressive downscaling on a given region, from the original resolution of a global model (order of 100 km) to the finer resolutions recommended for regional applications and impact studies. The use of increasingly high resolutions allows the system to develop its own dynamics at the intermediate scales that will feed the high-resolution model , as well as to avoid numerical instabilities and unreliable results.*

The authors spend much time in the abstract (where it is inappropriate) and in the introduction describing convection permitting (CP) models that need scale of <4 km.  But then, they use models that are not at CP scales, so why waste this space which should be better used describing what they did here.

> *3. REPLY:*
> *As one of the main results of the model configuration investigated is a model behavior very close to a convection-permitting configuration, we dedicated in depth discussion to the description of models at such resolutions, as background to show the advantages of our configuration at its highest resolution. As also reported in REPLY#14 we have demonstrated that the model at 5km in the current configuration behaves as a CP one.*

There is also a lot of talk about ECVs, but the comparisons are only to T2m and Precip., so why open discussion on ECVs here?

> *4. REPLY:*
> *We agree with the reviewer about mentioning ECVs, having dealt with only two of them. We'll remove ECVs from the text.*

The two model simulations WRF-15km and WRF-5km are compared against observations (with the ERA5 forcing) but there is less comparison and emphasis on the overlap region, the

only region of interest here since the case being made is what is the difference between 5km and 15km. In fact the experimental design is poorly thought out, Both 5km and 15km should have been run over the same domain, because making the 15 km domain so much larger changes the way the ALPS domain sees the boundaries. Better to run both for the same Italian region

**5. REPLY:**
See **REPLY#2** for a more detailed description of double nesting technique that should definitely clarify that it is not possible to run "5km and 15km over the same domain" as the 15 km run should provide BC to the inner 5 km domain and its continental size is thought in favour of satisfying reproduction of large scale condition over the region. The experiment is not merely an exercise of comparison of two different resolutions, but, as specified, a nesting strategy that then also allows to evaluate the benefits of high resolution that is desirable in a morphologically complex region such as the Mediterranean, where the benefits of moving toward the km-scale are largely demonstrated in literature.

L34 – "demands for high-res" is confusing use of English

**6. REPLY:**
Thanks for the comment: we will remove "for"

L35 – Really, so many regions over the globe are "critically prone to the impacts of local-scale and severe weather" Certainly the mountains are difficult to model, but not unique in tough weather.

**7. REPLY:**
As stated in the title of the article we are not studying any other region in the world but the Mediterranean, and how its climate has a negative impact on local communities. (See references)

L39 – No one is seriously running 200 km global climate models these days!

**8. REPLY:**
We'll better highlight in the revised text that several CMIP6 experiments run at this resolution (See Klaver et al., 2020).

L43-– what does "(120 divided by 20 km) mean? 6 km?

**9. REPLY:**
It means in the range between 120 km and 20 km, we will exchange the symbol for the periphrasis

L66- CP is not defined except in the abstract (does not count as text)

**10.REPLY:**
*We will move the definition of the acronym CP from the abstract to the text*

*L73-* "model uncertainties" or model errors?

**11.REPLY:**
*In this context we reported what stated in Fosser et al 2024*
*In the revised text we'll better remark the proposed reference*

These are some examples of why this paper is difficult to read and interpret.

**12.REPLY:**
*We'll improve the text readability accordingly to the previous replies*

Fig.5 – the description of this figure in the paper is contrary to what I see in it. The summertime dip in the 5km model is described as producing a much better seasonal fit than the 15km one. I see that the summer (months 6-7-8) in 5km is better, but for the rest of the year, it is still just like the 15km and overestimates the rain. "more realistic" is a marginal call. " better in summer" is OK

**13. REPLY:**
T*he description of Figure 5 states that the 5 km simulation reduces the summer bias from April to November. We will better explain Figure 5 and avoid to say "more realistic"*

Figure 6 is even more of a shock. The 5km run has NO cumulus rain, only annually uniform stratiform rain. The 15km run develops cumulus in summer, but the total (Fig 5) is far too large vs. observed. This indicates the your scale-aware convection is totally failing: as we go form 15km to 5km your cumulus convection parameterization shuts down.

*14.REPLY:*

*Figure 6 is one of the main points of the paper. In the revised text we'll remark that it demonstrates that the scale-aware parameterization is correctly working, at least for the experimental design here proposed. The 5 km model is run in the gray zone, this means that we did not switch off the convection parameterization (as it happens in CP models). Nevertheless, the G-F parameterization, due to the scale-awareness, dynamically manages the cumulus activity based on vertical mass-flux within a grid cell. Actually, the model behaves similarly to a CP one and benefits of an improved representation of the precipitation, at the same time saving computational time and data storage. We will make an effort to state more clearly the point in the text, as it seems prone to misunderstandings. The use of the term "stratiform" is actually misleading, as it's referred to the precipitation comes from the microphysics, due to both large and convective scale processes. The term "cumulus" here is referred to the parametrized contribution to the precipitation (i.e. it might be zero, or very small, if G-F is "switched off" and the convection is explicitly resolved).*

Figure 8 – The identical power spectrum for 5km and 15km tells me that both are doing the same thing because both have the same parameterization??

*15.REPLY:*

*In Fig.8 is reported the PDF of daily precipitation. However, it is unusual to have similar behavior for 5 km and 15 km. We already notice this point and it deserves further investigation. We will state more clearly this limit in the text, additional explanation and open questions will be added.*

Figures 12&13: If you want us to learn anything from this experiment, both should be plotted on the same grid so we can compare. As is, there is no useful information here.

*16. REPLY:*

*These figures actually refer to different variables for the same experiment. We suppose that the reviewer meant fig 12 and 14. In any case the set of figures from 12 to 16 display the climate signal for T and P for the parent (15 KM) and the child experiment (5 KM), respectively. Given the above explanation about double nesting, we think that it is clear why we do not plot them on the same grid. The useful information is that we are undoubtedly facing a warming and alterations of the precipitation patterns for the end of the century. In particular, the signal in precipitation for 5 km experiment is generally more statistically robust.*

*L381* – Conclusions:
    I would say this is a downscaling example, but certainly not a strategy.
    We have seen no evidence the your G-F cumulus scheme mimics a CP scheme.
      In fact it seems to shut down.
    The 5km runs improved a wet bias ONLY in summer, and they did that oddly (above)

How does this work show that your 5km or 15km runs are "adequate to provide the boundary conditions for" CP scale and further downscaling. This is simply not shown.

**17. REPLY:**
*We think that the above replies should have shed a light also on the conclusions and clarified the comments made by the reviewer.*

I am confused as to why this paper is a GMD paper, there is really no model development.

**18. REPLY:**
*We thank the reviewer for the comment which helps us clarify the scope of the paper. This paper deals with the description of a regional climate downscaling experiment. The development is not to be intended in model physics but in the setup. In particular, we recall here that one of the most important novelties of the paper is the use of CMIP6 forcing.*

An odd question: since you are forcing at the somewhat distant boundaries, do you not have synoptic climate variability in the interior, especially with regard to convection? Do you force internally? Therefore, would different ensembles produce different statistics?

**19. REPLY:**
*We think that the above replies should have shed a light also on this aspect.*

The code and data availability is very weak. Is the exact WRF version used here the "current version" with the github reference? Were any mods made? I agree it Is not the responsibility of these GMD authors to archive the ERA5 data. But where is the reference to the MPI-ESM data? (also on another archive). What is missing here is the actual values from the figures posted here. That is a useful minimum requirement. What code was used to process the data sets?

**20. REPLY:**
*We'll modify these aspects in the revised text accordingly to the GMD policies . We anticipate that the MPI-ESM, similarly to the other CMIP6 simulations, are freely available on the Earth System Grid Federation (ESGF).*

---

## Author Response (AR1)

**Reply to reviewers:  Manuscript Egusphere 2025-387**

"Impact of spatial resolution on CMIP6-driven Mediterranean climate simulations: a focus on precipitation distribution over Italy"

By: Maria Vittoria Struglia, Alessandro Anav, Marta Antonelli, Sandro Calmanti, Franco Catalano, Alessandro Dell'Aquila, Emanuela Pichelli, Giovanna Pisacane
https://egusphere.copernicus.org/preprints/2025/egusphere-2025-387/#discussion

**General remark**

In the revised version we changed the figures to be compliant with the GMD standards for readers with color vision deficiencies. However, for the sake of readability, we do not highlight the figure changes using the track change instrument.
Figure captions have been modified accordingly.

**Point by point reply to reviewers**

**RC1**
**Minor 1: If the authors could remark on how the donwscaling impacts the climate change signal of the global climate model over Italy, it would greatly enhance the manuscript's overall added value.**

**Author response**: We thank the reviewer for her/his positive feedback on the paper, and we addressed this specific point by adding three more figures that we are uploading as supplementary material. Figures S1 and S2 show, for the global driving model, the projected seasonal change of T2m and precipitation, respectively. We show the same panels of figures 14 and 15 of the manuscript over the common area of Italy and West-Med, to highlight the added value of the double nesting approach.
Although the warming signal shows a broad agreement in terms of intensity and spatial pattern between the driver and the high resolution experiment, some significant differences arise. There is a general reduction in the projected warming at 5km resolution with respect to the driver, especially during spring and summer, while in mountainous regions there is a larger warming that is not captured by the global model. Also land-sea contrasts appear better resolved in the high-resolution simulation.
As to the precipitation change signal, we note that while the driver projects a drier future for all seasons in the SSP2-4.5 and SS5-8.5 scenarios, the enhanced resolution experiment displays changes in the spatial patterns, exhibiting the intensification of wet anomalies and  with the signal even changing sign over extended areas  of the domain. This behaviour is already evident in the first downscaling at 15 km, and it is particularly evident during the fall season across coastal areas and over alpine topography, probably due to the improved representation of local interactions (sea-land, orographic forcings).
As regards the high-end tail of the precipitation distribution, Figure 18 shows that variations in the 99[th] percentile under future scenarios are more spatially variegated than the systematic increase exhibited by the global driver (Figure S3).

**Author changes in manuscript:** we added three figures in the supplementary material showing the climate signal of the global driver over Italy, and we added the following sentences in the manuscript:

L442 *"Projected warming is less intense compared to the results of the global forcing (Figure S1), especially in JJA and SON. Furthermore, the results at 5 km resolution clearly evidence larger warming over mountainous areas in JJA and SON for SSP2-4.5 and SSP5-8.5, which is not reproduced by the global model."*

L466 *"Interestingly, precipitation climate change signal in the downscaled projection is quite different from results by the global driver (Figure S2) and even of opposite sign over some areas in the shoulder seasons (MAM and SON). In particular, while in these seasons the global model projects an overall drying signal for SSP2-4.5 and SSP5-8.5, the regional model projects significant increase of mean seasonal precipitation in particular over the Alps and North-East in MAM and North Italy in SON. It is worth to note the change of signal at both levels of nesting (15 and 5km, Figures 13 and 15) in the fall season across the western-mediterranean coastal areas and over alpine topography, probably due to the improved representation of local interactions (sea-land, orographic forcings)."*

L516 *"In contrast to the complex spatial patterns observed in experiments D01 and D02, the global climate model MPI-ESM1-2-HR exhibits a more spatially uniform increase in the 99th percentile of daily rainfall (Figure 19), with no apparent relationship to local orography and considerably larger overall changes. The comparison between Figures 17–18 and S3 highlights a substantial improvement in the representation of local extreme events achieved through high-resolution downscaling. Notably, as also seen for changes in mean seasonal rainfall (Figure 15), the accurate depiction of projected changes in local extremes is strongly influenced by the representation of interactions with local orography, which plays a key role in constraining the spatial distribution of intense precipitation events."*

**Minor 2: Please add the colorbar units in figures 12-15**

**Author response**: we added the units in the captions

**Author changes in manuscript:** we changed the captions. The new captions are

*Figure 12: Projections of temperature change at the end of the century in the multi-scenario experiments – 15km SSPs ( °C)*

*Figure 13: Projections of precipitation change at the end of the century in the multi-scenario experiments 15km-SSPs (mm/day)*

*Figure 14 Projections of temperature change at the end of the century in the multi-scenario experiments – 5km SSPs ( °C)*

*Figure 15: Projections of precipitation change at the end of the century in the multi-scenario experiments 5km-SSPs (mm/day)*

**RC2**

**Authors' general response to the comments of Reviewer #2**: we thank the reviewer for his/her feedback and, as already stated in the first round of reply during the discussion phase, we revised the paper in order to clarify those aspects that may have be too synthetically described, i.e. the simulation protocol, or not fully appreciated, as the *quasi-convection permitting* configuration run in the grey zone. In the revised version of the paper, we expand section 2 in order to include a more thorough description of the numerical protocol. We also highlight one of the most important aspects of this work, i.e. the results obtained with a simulation in the grey zone for the convection representation (4-10 km), as well as its bearings on the optimal design of numerical simulations at such grid step range.
Circumstantiated comments will be dealt with as separated items in the following of this response.

**Comment 1: Please remove CMIP6 from the title**

**Author response:** We acknowledge that the current title might suggest expectations greater than those we are dealing with. However, the results shown in this paper are among the very first obtained when downscaling a last-generation CMIP6 global model and this is a distinctive feature of our work. We propose a new title that keeps the reference to CMIP6 while eliminating unclarity

**Author changes in manuscript:** We change the current title with *"Impact of spatial resolution on multi-scenario WRF-ARW simulations driven by the CMIP6 MPI-ESM1-2-HR global model: a focus on precipitation distribution over Italy"*.

**Comment 2: Basically, this is a paper not about model development, but about the simple application of an off-the-shelf WRF model run at two resolutions.**

**Author response:** We do not simply run an off the-shelf model at two resolutions. We applied and validated a double nesting approach to perform a progressive downscaling tailored on the Italian and West-Med region that allows to enhance the resolution of climate projections from the 100 km of the global driver to the fine resolution recommended for regional applications and impact studies. In addition, the second nesting level is in the so-called "gray zone" of convection representation, which is little or not at all studied and documented in literature especially at the climate scale, and which, in the final configuration tested, verified and chosen for our study, offers new application perspectives in a computation-saving perspective while not relinquishing the high resolution needed for the study of extreme events of the precipitation and its sub-daily scale, which is cutting-edge of applications in the field of climate modelling over the complex area of the Mediterranean (Lucas-Picher et al., 2021, https://doi.org/10.1002/wcc.731). We expanded section 2 with a detailed description of the methodology used and we added some references, not exhaustive but representative, of both the regional dynamical downscaling techniques and a couple of applications of the technique in other regions than the Mediterranean one.

**Author changes in manuscript:**

**Line 141:** We added the following paragraph

*The double nesting approach consists in performing regional downscaling of a global simulation by use of two domains with increasing spatial resolution: the parent simulation, at an intermediate resolution between the global model one and the finest one, provides initial and boundary conditions to the finest resolution experiment at the innermost level (i.e. child simulation). This approach is widely used to gradually enhance the grid resolution over a region of interest and it is demonstrated, particularly useful over complex morphology and orography regions (Im et al, 2006, Ji and Kang, 2013); it is an extension over multiple levels of nesting of the widely used regionalization technique based on the dynamical downscaling with a Regional Climate Model (Giorgi et al., 2001 and Giorgi, 2019). This approach allows to perform a progressive downscaling on a given region, from the original resolution of a global model (order of 100 km) to the finer resolutions recommended for regional applications and impact studies, while better dealing with the local forcings and interactions (e.g. complex topography, coastlines and land use). The use of increasing resolutions allows the system to develop its own dynamics at the intermediate scales that will feed the high-resolution model, as well as to avoid numerical instabilities and unreliable results.*

**Line 152:** We modified the sentence, the new version is

*In the current work the parent simulation (D01 in Fig.1) has a horizontal grid-step of 15 km, while the innermost nested domain D02 has a resolution of 5 km.*

**Comment 3: The authors spend much time in the abstract (where it is inappropriate) and in the introduction describing convection permitting (CP) models that need scale of <4 km. But then, they use models that are not at CP scales, so why waste this space which should be better used describing what they did here.**

**Author response** We thank the reviewer for the suggestion. As one of the main results of the model configuration investigated is a model behaviour very close to a convection-permitting configuration, we dedicated in depth discussion to the description of models at such resolutions, as background to show the advantages of our configuration at its highest resolution. The results presented in this manuscript suggest that the model at 5km in the current configuration behaves similarly to a CP one.
With this in mind, we think that it is better not to modify the abstract, as we are just mentioning one of the results of the paper.
By following reviewer's recommendation, we have extended the introduction to better explain what we did in the paper and the motivations to explore the grey zone, although with due caution.

**Author changes in manuscript**

**Line 86:** We modified the sentence, the new version is

*On the other hand, regional climate simulations typically span several decades and comparatively large domains and can still prove very expensive in terms of computational*

*resources (Fuhrer et al. 2018), even limiting grid spacing at the upper edge of the so-called gray zone.*

**Line 101:** We added the following paragraph as well as the references within.

*However, due to the computational effort required to produce climate simulations at convection permitting scales, compromises have so far been made on both the computational domain and the time length of the simulations (see for protocol in ex. Coppola et al, 2020). In order to produce sufficiently robust statistics for use in national risk assessment plans, both of these limitations must therefore be overcome. One possible compromise is to venture into the so-called gray zone for the convective schemes. This choice may enable to cover domains of regional interest and still allow for simulations, whether reference or projections, that are long enough to give some robustness to the statistics of extreme events as well. In any case, one should be cautious in order not to run into ambiguous results due to the use of parameterizations that are not suitable for the scale at which one is working (Prein et al., 2015). These cautions should be extended not only to the parameterization of convection but also to the related ones of microphysics and planetary boundary layer (Jeworreck et al., 2019). The Weather Research and Forecasting model with the Advanced Research core (AR-WRF, Skamarock et al., 2008) provides a wide suite of parameterizations to choose from, including the scale-aware ones that have proved effective in managing the transition to more resolved scale in many tests (Liu et al., 2011, Jeworreck et al., 2019, Park et al., 2024).*

**Line 118** We modified the sentence, the new version is

*The Weather Research and Forecasting model (WRF) is used, first over the EURO-CORDEX domain, at a horizontal spatial resolution of 15 km, and then over an inner domain at a finer grid covering the whole of Italy and extending to the north-western Mediterranean at a resolution of 5 km, i.e. within a grid-step range in the gray zone for cumulus representation. Future scenarios continuously extend from present time up to 2100.*

**Comment 4: There is also a lot of talk about ECVs, but the comparisons are only to T2m and Precip., so why open discussion on ECVs here?**

**Author response** We agree with the reviewer about mentioning ECVs, having dealt with only two of them. We'll remove ECVs from the text.

**Author changes in manuscript**

**Line 22** We modified the sentence, the new version is

*Analyses of air temperature and precipitation are presented,..*

**Line 248** We modified the sentence, the new version is

*These variables are chosen because they affect ...*

**Comment 5: The two model simulations WRF-15km and WRF-5km are compared against observations (with the ERA5 forcing) but there is less comparison and emphasis on the overlap region, the only region of interest here since the case being made is what is the**

**difference between 5km and 15km. In fact the experimental design is poorly thought out, Both 5km and 15km should have been run over the same domain, because making the 15 km domain so much larger changes the way the ALPS domain sees the boundaries. Better to run both for the same Italian region**

**Author response** There has been probably a misunderstanding regarding the methodology and we believe the current version of the manuscript has been improved in the description of the aim and experimental design. Please see **Response #2** for a more detailed description of double nesting technique. In particular, we recall here that is not possible to run "5km and 15km over the same domain" as the 15 km parent domain provides BC to the inner 5 km domain and its continental size is thought in favour of satisfying reproduction of large scale condition over the region. The objective of the study is not the comparison of two different resolutions, but, as better described in the revised manuscript, a nesting strategy that allows to evaluate the benefits of a gradual dynamical downscaling up to the resolution fine enough as desirable in a morphologically complex region such as the Mediterranean, where the benefits of moving toward the km-scale are largely demonstrated in literature.

**Author changes in manuscript** *See **Author changes #2***

**Comment 6: L34 – "demands for high-res" is confusing use of English**

**Author response** Thanks for the comment: we will remove "for"

**Author changes in manuscript**

**New L41** We modified the sentence, the new version is

...*which inherently demands high-resolution analyses*

**Comment 7: L35 – Really, so many regions over the globe are "critically prone to the impacts of local-scale and severe weather" Certainly the mountains are difficult to model, but not unique in tough weather.**

**Author response** As stated in the title of the article we are not studying any other region in the world but the Mediterranean, and how its climate has a negative impact on local communities. The reference to the Mediterranean is not meant to minimize the fact that extreme events occur in other parts of the world as well, but serves to contextualize the rationale for our work, which was carried out in the context of projects dedicated to studying the impacts of climate change on the Mediterranean region and the possible adaptation strategies. Anyway, the Mediterranean is a region intrinsically difficult to model at the mid-latitudes due to the presence of the sea close to the steep orography of the Alps and Apennines. The proposed references of Rotunno and Houze, 2007 and Ducrocq et al, 2014 well clarify this point. Unfortunately, heavy rainfall events have in the past generated flash floods and/or triggered ground instabilities with major consequences on socioeconomic sectors and even casualties in several regions of Italy. The references cited (Rebora et al., 2013; Arrighi and Domeneghetti,2024) are just a couple of examples referring to recent events occurred in the Liguria and Emilia-Romagna regions.

**Author changes in manuscript**

No change has been made in the manuscript

**Comment 8: L39 – No one is seriously running 200 km global climate models these days!**

**Author response**

Actually, several CMIP6 experiments run at this resolution. In the text we refer to the paper of Klaver et al., 2019 that compares nominal and effective resolution across several state-of-the-art global models that participate in the CMIP6 experiment. Table 1 of this paper shows nominal and effective resolution of 13 models, that range from 38.2 km to 217 km and from 182 km to 625 km, respectively. We are aware that some research groups have achieved extremely high resolutions even with global models, which in any case have been cited few lines after in the same sentence (HighResMIP, Haarsma et al., 2016), but for the current work our reference is the CMIP6 experiment.

**Author changes in manuscript**

**New L47** We modified the sentence, the new version is

*CMIP6 state of the art global projections typically have a nominal horizontal grid resolution that roughly ranges from 38 km to 200km, corresponding to a three/five-time larger effective resolution (Klaver et al., 2020).*

**Comment 9: L43-– what does "(120 divided by 20 km) mean? 6 km?**

**Author response** It means in the range between 120 km and 20 km, we exchanged the symbol for the periphrasis

**Author changes in manuscript**

**New L51** We modified the sentence, the new version is

*…resolution (in the range from 120 km to 20 km)…*

**Comment 10: L66- CP is not defined except in the abstract (does not count as text)**

**Author response** The acronym has been added in the text

**Author changes in manuscript**

**New L78** We modified the sentence, the new version is

*The added value of resolving the Convection Permitting (CP) scale..*

**Comment 11: L73 - "model uncertainties" or model errors?**

**Author response** Model uncertainties. The cited article (Fosser et al 2024) deals with the evaluation of model uncertainties in a CPM multi-model ensemble, basing on the CORDEX Flagship Pilot Study project, with respect to the uncertainties of its driving convection-parametrized RCM ensemble. One of the main findings of the paper is that the use of CPMs allows to reduce model uncertainty, for both the present climate and the future changes, due to the more realistic representation of local dynamical processes.

This citation is important in this part of the introduction as we are describing pros (robust high spatial and temporal resolution climate information, reduction of uncertainties, better representation of extremes...)  and cons (computational effort, data storage, small domain and/or time slices) of CPMs.  The new simulation that we are presenting in this paper partially overcomes some of the cons while preserving the pros, at least for the precipitation statistics.

**Author changes in manuscript** We do not modify this specific sentence, but we think that the changes made in the introduction and already reported in **response #3** may clarify also this aspect

**Comment 12: Fig.5 – the description of this figure in the paper is contrary to what I see in it.  The summertime dip in the 5km model is described as producing a much better seasonal fit than the 15km one.  I see that the summer (months 6-7-8) in 5km is better, but for the rest of the year, it is still just like the 15km and overestimates the rain.  "more realistic" is a marginal call.  " better in summer" is OK**

**Author response** The description of Figure 5 states that the 5 km simulation reduces the wet bias from May to October, mostly evident in summer. We have better explained Figure 5 and avoided to say "more realistic".

**Author changes in manuscript**

**New Line 237** We modified the sentence, the new version follows**.** We also added symbols in Fig.5 to improve readability by readers with colour vision deficiencies.

*"The 15km-Hindcast experiment (red line/diamonds symbols) has a wet bias with respect to all the reference datasets throughout the year of the order of 1 mm/day and 2 mm/day compared to ERA5(arrow symbols) and E-obs (square symbols), respectively. The 5km-Hindcast curve (orange line/dot symbols) reproduces a better seasonal variability characterized by the two relative maxima during spring and autumn, and it is closer to the observations especially in summer, clearly reducing the biases compared to its 15km-Hindcast driver within May and October."*

**Comment 13:  Figure 6 is even more of a shock.  The 5km run has NO cumulus rain, only annually uniform stratiform rain.  The 15km run develops cumulus in summer, but the total (Fig 5) is far too large vs. observed.  This indicates the your scale-aware convection is totally failing:  as we go form 15km to 5km your cumulus convection parameterization shuts down.**

**Author response** We thank the referee for the comment which allows us to better clarify one of the main points of the paper. In the revised text we remark that Figure 6 demonstrates that

the scale-aware parameterization is correctly working, at least for the experimental design here proposed. The 5 km model is run in the gray zone, this means that we did not switch off the convection parameterization (as it happens in CP models). Nevertheless, the G-F parameterization, due to the scale-awareness, dynamically manages the cumulus activity based on vertical mass-flux within a grid cell. Actually, the model behaves similarly to a CP one and benefits of an improved representation of the precipitation, at the same time saving computational time and data storage. We made an effort to state more clearly this point in the text, as it seems prone to misunderstandings. We also avoid the use of the term "stratiform", which is actually misleading, as it's referred to the precipitation that comes from the microphysics, due to both large and convective scale explicit processes. The term "cumulus" was referred to the parametrized contribution to the precipitation (i.e. it might be zero, or very small, if G-F is "switched off" and the convection is explicitly resolved).

**Author changes in manuscript**

**New L307 and following** We rephrased the paragraph and added some explanations. The new paragraph reads:

*"In Figure 6 we analyzed in detail how the two contributions to the total daily precipitation are produced by the schemes of cumulus and microphysics in the hindcast simulations. In the 15km-Hindcast (red), the seasonal cycles of the precipitation parameterized by the convection scheme (solid line) and the component coming from the microphysics scheme (dashed) and explicitly resolved have maxima of the same order of magnitude, although in different seasons, as expected. By comparison with the 5km-Hindcast curves (orange) and keeping in mind the results in Figure 5, we can speculate that an overestimation of the contribution coming from the cumulus parameterization during summer is the cause for the wet bias in the 15km-Hindcast. On the other hand, the contribution of the convective parametrized precipitation in the 5km-Hindcast (orange solid curve) is one order of magnitude lower than the one at coarse resolution for every month of the year, thus signifying that most of the precipitation, either large scale or convective, is explicitly resolved. This suggests that the model in the gray-zone, due to the scale-aware behavior implemented in the Grell-Freitas scheme (Freitas et al, 2021), mimics a convection permitting model, smoothing the transition from sub-grid (cumulus) to resolved-scale (microphysics) with increasing resolution (Jeworrek et al., 2019)."*

**Comment 14: Figure 8 – The identical power spectrum for 5km and 15km tells me that both are doing the same thing because both have the same parameterization??**

**Author response.** We agree that it is not obvious to have similar behavior for 5 km and 15 km in the PdF. The two RCMs have indeed the same parameterization, although it is working differently at the two resolutions due to its scale awareness. As discussed in the previous paragraphs, the high-resolution experiment corrects the anomalous wet bias of the parent simulation, which is due, at least in the summer season, to an overrepresentation of the convective events that might explain the anomalous distribution of the extreme events in the 15 km experiment. As highlighted at line 374 of the paper, we can consider a sort of an anomalous behavior the long tail of the 15 km hindcast (Pichelli et al., 2021 in the supplementary material).

**Author changes in manuscript** We do not modify this specific sentence, but we think that the changes reported in **response #13** may clarify also this aspect

**Comment 15: Figures 12&13: If you want us to learn anything from this experiment, both should be plotted on the same grid so we can compare. As is, there is no useful information here.**

**Author response** These figures actually refer to different variables for the same experiment. We suppose that the reviewer meant fig 12 and 14. In any case the set of figures from 12 to 16 display the climate signal for T and P for the parent (15 KM) and the child experiment (5 KM), respectively. Please refer to reply to comment #2 for an explanation of double nesting methodology where we explain why we do not plot them on the same domain area. The useful information in these figures is a detailed picture of projected warming and alterations of the precipitation patterns for the end of the century over the Italian peninsula, every simulations relatively to its own grid. In particular, the signal in precipitation for 5 km experiment is generally more statistically robust than in the coarse resolution of the GCM and also compared to the 15 km resolution domain. Torma et al., 2015 have largely demonstrated that if any added value is produced by enhancing the resolution with a dynamical downscaling, such added value is maintained even in case of re-gridding at a coarser resolution.

**Author changes in manuscript** We do not modify this specific sentence, but we think that the changes reported in **response #2** may clarify also this aspect

**Comment 16:** *L381* **– Conclusions: I would say this is a downscaling example, but certainly not a strategy. We have seen no evidence the your G-F cumulus scheme mimics a CP scheme. In fact it seems to shut down. The 5km runs improved a wet bias ONLY in summer, and they did that oddly (above). How does this work show that your 5km or 15km runs are "adequate to provide the boundary conditions for" CP scale and further downscaling. This is simply not shown.**

**Author response** We think that the above replies should have shed light also on the conclusions and clarified the comments made by the reviewer. Anyway, we applied some changes in the paragraph, and we avoid to use the term "adequate" leaving space for potential use of our data for further dynamical downscaling toward convection permitting scale.

**Author changes in manuscript**

**L547** rephrased as *"A dynamical downscaling strategy, from global model scale to a regional scale laying within the grey zone (4-10km) for the representation of the convection, has been performed and validated to produce multi-scenario regional climate simulations with the atmospheric model WRF for the Mediterranean region"*

**L554** *"In this work, we explore a grid-step (5km) at the lower edge of the gray-zone range, …"*

**L573** *"Finally, let us remark that both the intermediate and high-resolution simulations are potentially usable as boundary conditions for convection-permitting scale (finer than 4 km) further downscaling."*

**Comment 17: I am confused as to why this paper is a GMD paper, there is really no model development.**

**Author response** We thank the reviewer for the comment which helps us clarify the scope of the paper. This paper deals with the description of a regional climate downscaling experiment up to horizontal scales that fall in the gray-zone. The development is to be intended on one hand in the experimental setup that uses the double nesting approach, on the other hand in the exploration of the gray zone which is usually avoided, so poorly or not documented in literature. In addition, we recall here that another important novelty of the paper is the use of CMIP6 forcing. All these aspects have been clarified in the revised version of the manuscript

**Author changes in manuscript** Many changes have been done to the manuscript, to better highlight the novelties of the work. See for example the **reply to Comment 2**

**Comment 18:** An odd question: since you are forcing at the somewhat distant boundaries, do you not have synoptic climate variability in the interior, especially with regard to convection? Do you force internally? Therefore, would different ensembles produce different statistics?

**Author response** We think that the above replies should have shed light also on this aspect.

**Author changes in manuscript** Changes to the manuscript were not applicable, as to this specific question of the reviewer.

**Comment 19: The code and data availability is very weak. Is the exact WRF version used here the "current version" with the github reference? Were any mods made? I agree it Is not the responsibility of these GMD authors to archive the ERA5 data. But where is the reference to the MPI-ESM data? (also on another archive). What is missing here is the actual values from the figures posted here. That is a useful minimum requirement. What code was used to process the data sets?**

**Author response** No modifications were done to the cited WRF version. The chosen parameterizations were listed in table 1. The MPI-ESM data are freely available on the Earth System Grid Federation (ESGF). The analysis of the results were done using different tools (e.g. shell coding, python coding, R coding, ncl coding), for standard statistics and graphic purposes, and there is not a unique code for the data processing. Anyway the different scripts as well as model output data can be made available upon request, as already stated in the section.

**Author changes in manuscript** We added the following sentence

**L601** *The MPI-ESM data are freely available on the Earth System Grid Federation (ESGF).*